# A GRAPH-BASED GLOBAL OPTIMIZATION FRAMEWORK FOR PROBLEMS WITH NONCONVEX NORM CONSTRAINTS AND PENALTY FUNCTIONS

## ABSTRACT

Optimization problems with norm-bounding constraints appear in various applications, from portfolio optimization to machine learning, feature selection, and beyond. A widely used variant of these problems relaxes the norm-bounding constraint through Lagrangian relaxation and moves it to the objective function as a form of penalty or regularization term. A challenging class of these models uses the zero-norm function to induce sparsity in statistical parameter estimation models. Most existing exact solution methods for these problems use additional binary variables together with artificial bounds on variables to formulate them as a mixed-integer program in a higher dimension, which is then solved by off-the-shelf solvers. Other exact methods utilize specific structural properties of the objective function to solve certain variants of these problems, making them non-generalizable to other problems with different structures. An alternative approach employs nonconvex penalties with desirable statistical properties, which are solved using heuristic or local methods due to the structural complexity of those terms. In this paper, we develop a novel graph-based method to globally solve optimization problems that contain a generalization of norm-bounding constraints. This includes standard $\ell_p$-norms for $p \in [0, \infty)$ as well as nonconvex penalty terms, such as SCAD and MCP, as special cases. Our method uses decision diagrams to build strong convex relaxations for these constraints in the original space of variables without the need to introduce additional auxiliary variables or impose artificial variable bounds. We show that the resulting convexification method, when incorporated into a spatial branch-and-cut framework, converges to the global optimal value of the problem under mild conditions. To demonstrate the capabilities of the proposed framework, we conduct preliminary computational experiments on benchmark sparse linear regression problems with challenging nonconvex penalty terms that cannot be modeled or solved by existing global solvers.

## 1 INTRODUCTION

Norm-bounding constraints are often used in optimization problems to improve model stability and guide the search towards solutions with desirable properties. For example, in machine learning, norm-bounding constraints are imposed as a form of regularization to reduce overfitting and induce sparsity in feature selection models Hastie et al. (2009). Other uses of these constraints include improving the numerical stability of optimization algorithms Nocedal & Wright (2006), controlling the complexity of the solution space Bertsekas (1999), and enhancing the interpretability of parameter estimation Zou & Hastie (2005). While norm-bounding constraints defined by $\ell_p$-norm for $p \geq 1$ fall within the class of convex programs, making them amenable to solution techniques from numerical optimization Nocedal & Wright (2006) and convex optimization Bertsekas (2015) literature, constraints involving $\ell_p$-norm for $p \in [0, 1)$ or their nonconvex proxies belong to the class of mixed-integer nonlinear (nonconvex) programs (MINLPs), posing greater challenges for global solution approaches.

The challenge to solve problems with $\ell_p$-norm for $p \in (0, 1)$ stems from the nonconvexity of this function, which necessitates the addition of new constraints and auxiliary variables to decompose the function into smaller terms with simpler structures. These terms are convexified separately through the so-called *factorable decomposition*, the predominant convexification technique in existing global solvers; see Bonami et al. (2012); Khajavirad et al. (2014) for a detailed account. In contrast, the challenge in solving problems with $\ell_0$-norm, also referred to as *best-subset selection* problem, is due to the discontinuity of the norm function, which necessitates the introduction of new binary variables and (often) artificial bounds on variables to reformulate it as a mixed-integer program (MIP) that can be solved by existing MIP solvers Bertsimas et al. (2016); Dedieu et al. (2021). Other exact methods for solving these problems are highly tailored to exploit the specific structural properties of the objective function, often making them unsuited for problems with different structures Bertsimas & Parys (2020); Hazimeh et al. (2022); Xie & Deng (2020). One of the most common applications of the best-subset selection problem is in sparse parameter estimation, where the goal is to limit the number of nonzero parameters Hastie et al. (2015); Pilanci et al. (2015). Other applications of problems with $\ell_0$-norm constraints include compressive sensing, metabolic engineering, and portfolio selection, among others; see Jain (2010) and references therein for an exposition on these applications. Perhaps the most challenging class of these problems, from a global optimization perspective, involves nonconvex penalty functions that exhibit desirable statistical properties, such as variable selection consistency and unbiasedness Fan & Li (2001); Zhang & Zhang (2012). The complexity of these problems is partly attributed to the algebraic form of their penalty functions, which are often difficult to model as standard mathematical programs that can be solved by a global solver. For example, the smoothly clipped absolute deviation (SCAD) Fan & Li (2001) and the minimax concave penalty (MCP) Zhang (2010) functions include integration in their definition, an operator that is inadmissible in existing global solvers such as BARON Tawarmalani & Sahinidis (2005). As a result, the existing optimization approaches for these problems mainly consist of heuristic and local methods Mazumder et al. (2011); Zou & Li (2008). Various works in the literature have studied the favorable properties of global solutions of these statistical models, which are often not achievable by local methods or heuristic approaches Fletcher et al. (2009); Hazimeh & Mazumder (2020); Wainwright (2009). This advocates for the need to develop global optimization methods that, despite being computationally less efficient and scalable compared to heuristic or local methods, can provide deeper insights into the true statistical characteristics of optimal estimators across various models. Consequently, these insights can facilitate the development of new models endowed with more desirable properties; see the discussion in Fan & Li (2001) for an example of the process to design a new model.

In this paper, we introduce a novel global optimization framework for a generalized class of norm-bounding constraints based on the concept of decision diagrams (DDs), which are special-structured graphs that draw out latent variable interactions in the constraints of the MINLP models. We refer the reader to Section 2 for a brief background on DDs, to Bergman et al. (2016) for an introduction to DDs, and to Castro et al. (2022) for a recent survey on DD-based optimization. One of the most prominent advantages of DD-based solution methods compared to alternative approaches is the ability of DDs to model complex constraint forms, such as nonconvex, nonsmooth, black-box, etc., while capturing their underlying data structure through a graphical representation. In this work, we exploit these properties of DDs to design a framework to globally solve optimization problems that include norm-bounding constraints, which encompass a variety of challenging nonconvex structures.

The main contribution of this paper is the development of a global solution framework through the lens of DDs, which has the following advantages over existing global optimization tools and techniques.

   (i) Our framework is applicable to a generalized definition of the norm functions that includes $\ell_p$-norm with $p \in [0, \infty)$, as well as nonconvex penalty terms, such as SCAD and MCP, as special cases.

   (ii) The devised method guarantees convergence to a global optimal solution of the underlying optimization model under mild conditions.

   (iii) The proposed approach provides a unified framework to handle different norm and penalty types, ranging from convex to nonconvex to discontinuous, unlike existing techniques that employ a different tailored approach to model and solve each norm and penalty type.

(iv) Our framework models the norm-bounding constraints and solves the associated optimization problem in the original space of variables without the need to introduce additional auxiliary variables, unlike conventional approaches in the literature and global solvers that require the introduction of new auxiliary variables for $\ell_p$-norms for $p \in [0, 1)$.

(v) Our approach can model and solve nonconvex penalty functions that contain irregular operators, such as the integral in SCAD and MCP functions, which are considered intractable in state-of-the-art global solvers, thereby providing the first global solution method for such structures.

(vi) The developed framework does not require artificial large bounds on the variables, which are commonly imposed when modeling $\ell_0$-norms as a MIP.

(vii) Our algorithms can be easily incorporated into solution methods for optimization problems with a general form of objective function and constraints, whereas the majority of existing frameworks designed for $\ell_0$-norm or nonconvex penalty terms heavily rely on the problem-specific properties of the objective function and thus are not generalizable to problems that have different objective functions and constraints.

(viii) Our approach can be used to model the regularized variant of the problems where the norm functions are moved to the objective function through Lagrangian relaxation and treated as a weighted penalty.

**Notation.** Vectors of dimension $n \in \mathbb{N}$ are denoted by bold letters such as $\boldsymbol{cx}$, and the non-negative orthant in dimension $n$ is referred to as $\mathbb{R}^n_+$. We define $[n] := \{1, 2, \ldots, n\}$. We refer to the convex hull of a set $P \subseteq \mathbb{R}^n$ by $\mathrm{conv}(P)$. For a sequence $\{t_1, t_2, \ldots\}$ of real-valued numbers, we refer to its limit inferior as $\liminf_{n \to \infty} t_n$. For a sequence $\{P_1, P_2, \ldots\}$ of monotone non-increasing sets in $\mathbb{R}^n$, i.e. $P_1 \supseteq P_2 \supseteq \ldots$, we denote by $\{P_j\} \downarrow \tilde{P}$ the fact that this sequence converges (in the Hausdorff sense) to the set $\tilde{P} \subseteq \mathbb{R}^n$. For $x \in \mathbb{R}$, we define $(x)_+ := \max\{0, x\}$.

## 2 BACKGROUND ON DDS

In this section, we present basic definitions and results relevant to our DD analysis. A DD $\mathcal{D}$ is a directed acyclic graph denoted by the triple $(\mathcal{U}, \mathcal{A}, l)$ where $\mathcal{U}$ is a node set, $\mathcal{A}$ is an arc set, and $l : \mathcal{A} \to \mathbb{R}$ is an arc label mapping for the graph components. This DD is composed of $n \in \mathbb{N}$ arc layers $\mathcal{A}_1, \mathcal{A}_2, \ldots, \mathcal{A}_n$, and $n + 1$ node layers $\mathcal{U}_1, \mathcal{U}_2, \ldots, \mathcal{U}_{n+1}$. The node layers $\mathcal{U}_1$ and $\mathcal{U}_{n+1}$ contain the root $r$ and the terminal $t$, respectively. In any arc layer $j \in [n] := \{1, 2, \ldots, n\}$, an arc $(u, v) \in \mathcal{A}_j$ is directed from the tail node $u \in \mathcal{U}_j$ to the head node $v \in \mathcal{U}_{j+1}$. The *width* of $\mathcal{D}$ is defined as the maximum number of nodes at any node layer $\mathcal{U}_j$. DDs have been traditionally used to model a bounded integer set $\mathcal{P} \subseteq \mathbb{Z}^n$ such that each $r$-$t$ arc-sequence (path) of the form $(a_1, \ldots, a_n) \in \mathcal{A}_1 \times \ldots \times \mathcal{A}_n$ encodes a point $\boldsymbol{cx} \in \mathcal{P}$ where $l(a_j) = x_j$ for $j \in [n]$, that is $\boldsymbol{cx}$ is an $n$-dimensional point in $\mathcal{P}$ whose $j$-th coordinate is equal to the label value $l(a_j)$ of arc the $a_j$. For such a DD, we have $\mathcal{P} = \mathrm{Sol}(\mathcal{D})$, where $\mathrm{Sol}(\mathcal{D})$ represents the set of all $r$-$t$ paths.

The graphical property of DDs can be exploited to optimize an objective function over a discrete set $\mathcal{P}$. To this end, DD arcs are weighted in such a way that the weight of an $r$-$t$ path, obtained by the summation of the weights of its arcs, encoding a solution $\boldsymbol{cx} \in \mathcal{P}$ equals the objective function value evaluated at $\boldsymbol{cx}$. Then, a shortest (resp. longest) $r$-$t$ path for the underlying minimization (resp. maximization) problem is found through a polynomial-time algorithm to obtain an optimal solution of the underlying integer program.

If there is a one-to-one correspondence between the $r$-$t$ paths of the DD and the discrete set $\mathcal{P}$, we say that the DD is *exact*. It is clear that the construction of an exact DD is computationally prohibitive due to the exponential growth rate of its size. To address this difficulty, the concept of *relaxed* and *restricted* DDs were developed in the literature to keep the size of DDs manageable. In a relaxed DD, if the number of nodes in a layer exceeds a predetermined *width limit*, a subset of nodes are merged into one node to reduce the number of nodes in that layer and thereby satisfy the width limit. This node-merging operation is performed in such a way that all feasible solutions of the exact DD are maintained, while some new infeasible solutions might be created during the merging process. Optimization over this relaxed DD provides a dual bound to the optimal solution of the original integer program. In a restricted DD, the collection of all $r$-$t$ paths of the DD encode a subset of the feasible solutions of the exact DD that satisfies the prescribed width limit. Optimization over this

restricted DD provides a primal bound to the optimal solution of the original integer program. The restricted and relaxed DDs can be successively refined through a branch-and-bound framework until their primal and dual bounds converge to the optimal value of the integer program.

As outlined above, DDs have been traditionally used to model and solve discrete optimization problems. Recently, through a series of works Davarnia (2021); Salemi & Davarnia (2022; 2023), the extension of DD-based optimization to mixed-integer programs was proposed together with applications in new domains, from energy systems to transportation, that include a mixture of discrete and continuous variables. In this paper, we make use of some of the methods developed in those works to build a global solution framework for optimization problems with norm-bounding constraints.

## 3 SCALE FUNCTION

In this section, we introduce the scale function as a generalization of well-known norm functions.

**Definition 1.** *For $cx \in \mathbb{R}^n$, define the "scale" function $\eta(cx) = \sum_{i \in N} \eta_i(x_i)$, where $\eta_i : \mathbb{R} \to \mathbb{R}_+$ is a real-valued univariate function such that*

(i) $\eta_i(x) = 0$ *if and only if $x = 0$,*

(ii) $\eta_i(x_1) \leq \eta_i(x_2)$ *for $0 \leq x_1 \leq x_2$ (monotone non-decreasing),*

(iii) $\eta_i(x_1) \geq \eta_i(x_2)$ *for $x_1 \leq x_2 \leq 0$ (monotone non-increasing).*

The definition of the scale function $\eta(cx)$ does not impose any assumption on the convexity, smoothness, or even continuity of the terms involved in the function, leading to a broad class of possible functional forms. In fact, as a special case, this function takes the form of the nonconvex penalty function outlined in Fan & Li (2001); Zhang (2010), which is commonly used as a regularization factor in the Lagrangian formulations of statistical estimation problems. Next, we show that two of the most prominent instances of such penalty functions, namely SCAD and MCP, are scale functions.

**Proposition 1.** *Consider the SCAD penalty function $\rho^{\text{SCAD}}(cx, c\lambda, c\gamma) = \sum_{i=1}^n \rho_i^{\text{SCAD}}(x_i, \lambda_i, \gamma_i)$ where $\lambda_i > 0$ and $\gamma_i > 2$, for $i \in [n]$, are degree of regularization and nonconvexity parameters, respectively, and $\rho_i^{\text{SCAD}}(x_i, \lambda_i, \gamma_i) = \lambda_i \int_0^{|x_i|} \min\{1, (\gamma_i - y/\lambda_i)_+/(\gamma_i - 1)\}dy$. Similarly, consider the MCP penalty function $\rho^{\text{MCP}}(cx, c\lambda, c\gamma) = \sum_{i=1}^n \rho_i^{\text{MCP}}(x_i, \lambda_i, \gamma_i)$ where $\lambda_i > 0$ and $\gamma_i > 0$ are degree of regularization and nonconvexity parameters, respectively, and $\rho_i^{\text{MCP}}(x_i, \lambda_i, \gamma_i) = \lambda_i \int_0^{|x_i|} (1 - y/(\lambda_i \gamma_i))_+ dy$. Then, $\rho^{\text{SCAD}}(cx, c\lambda, c\gamma)$ and $\rho^{\text{MCP}}(cx, c\lambda, c\gamma)$ are scale functions in $cx$.*

*Proof.* We show the result for $\rho^{\text{SCAD}}(cx, c\lambda, c\gamma)$ as the proof for $\rho^{\text{MCP}}(cx, c\lambda, c\gamma)$ follows from similar arguments. It is clear that $\rho_i^{\text{MCP}}(0, \lambda_i, \gamma_i) = 0$. Further, since $\min\{1, (\gamma_i - y/\lambda_i)_+/(\gamma_i - 1)\} \geq 0$ for $\gamma_i > 2$, we conclude that the integral function $\lambda_i \int_0^{|x_i|} \min\{1, (\gamma_i - y/\lambda_i)_+/(\gamma_i - 1)\}dy$ is non-decreasing over the interval $[0, \infty)$ for $|x_i|$ as $\lambda_i > 0$, proving the result. □

As a more familiar special case, we next show how the $\ell_p$-norm constraints for all $p \in [0, \infty)$ can be represented using scale functions.

**Proposition 2.** *Consider a norm-bounding constraint of the form $||cx||_p \leq \beta$ for some $\beta \geq 0$ and $p \in [0, \infty)$, where $||cx||_p$ denotes the $\ell_p$-norm. Then, this constraint can be written as $\eta(cx) \leq \bar{\beta}$ for a scale function $\eta(cx)$ such that*

(i) *if $p \in (0, \infty)$, then $\eta_i(x_i) = x_i^p$ and $\bar{\beta} = \beta^p$,*

(ii) *if $p = 0$, then $\eta_i(x_i) = 0$ for $x_i = 0$, and $\eta_i(x_i) = 1$ for $x_i \neq 0$, and $\bar{\beta} = \beta$.*

*Proof.*     (i) Using the definition $||cx||_p = \left(\sum_{i \in N} |x_i|^p\right)^{1/p}$ for $p \in (0, \infty)$, we can rewrite constraint $||cx||_p \leq \beta$ as $\sum_{i \in N} |x_i|^p \leq \beta^p = \bar{\beta}$ by raising both sides to the power of $p$. The left-hand-side of this inequality can be considered as $\eta(cx) = \sum_{i \in N} \eta_i(x_i)$ where $\eta_i(x_i) = |x_i|^p$. Since $\eta_i(x_i)$ satisfies all three conditions in Definition 1, we conclude that $\eta(cx)$ is a scale function.

(ii) Using the definition $||cx||_0 = \sum_{i \in N} \mathtt{I}(x_i)$ where $\mathtt{I}(x_i) = 0$ if $x_i = 0$, and $\mathtt{I}(x_i) = 1$ if $x_i \neq 0$, we can rewrite constraint $||cx||_0 \leq \beta$ as $\sum_{i \in N} \eta_i(x_i) \leq \beta$ where $\eta_i(x_i) = \mathtt{I}(x_i)$. Since $\eta_i(x_i)$ satisfies all three conditions in Definition 1, we conclude that $\eta(cx)$ is a scale function.

$\square$

We note that $\ell_\infty$-norm is excluded in the definition of scale function, as constraints of the form $||cx||_\infty \leq \beta$ can be broken into multiple separate constraints bounding the magnitude of each component, i.e., $|x_i| \leq \beta$ for $i \in [n]$.

## 4 GRAPH-BASED CONVEXIFICATION METHOD

In this section, we develop a novel convexification method for the feasible regions described by a scale function. We present these results for the case where the scale functions appear in the constraints of the optimization problem. The extension to the case where these constraints are relaxed and moved to the objective function as a penalty term follows from similar arguments. In the remainder of the paper, we refer to a constraint that imposes an upper bound on a scale function as a *norm-bounding* constraint.

### 4.1 DD-BASED RELAXATION

Define the feasible region of a norm-bounding constraint over the variable domains as $\mathcal{F} = \{cx \in \prod_{i=1}^n [\mathtt{l}^i, \mathtt{u}^i] \mid \eta(cx) \leq \beta\}$. For each $i \in [n]$, define $L_i$ to be the index set for sub-intervals $[\mathtt{l}_j^i, \mathtt{u}_j^i]$ for $j \in L_i$ that span the entire domain of variable $x_i$, i.e., $\bigcup_{j \in L_i}[\mathtt{l}_j^i, \mathtt{u}_j^i] = [\mathtt{l}^i, \mathtt{u}^i]$. Algorithm 1 gives a top-down procedure to construct a DD that provides a relaxation for the convex hull of $\mathcal{F}$, which is proven next.

---

**Algorithm 1:** Relaxed DD for a norm-bounding constraint

**Data:** Set $\mathcal{F} = \{cx \in \prod_{i=1}^n [\mathtt{l}^i, \mathtt{u}^i] \mid \eta(cx) \leq \beta\}$, and the domain partitioning intervals $L_i$ for $i \in [n]$
**Result:** A DD $\mathcal{D} = (\mathcal{U}, \mathcal{A}, l(.))$

1 create the root node $r$ in the node layer $\mathcal{U}_1$ with state value $s(r) = 0$
2 create the terminal node $t$ in the node layer $\mathcal{U}_{n+1}$
3 **forall** $i \in [n-1]$, $u \in \mathcal{U}_i$, $j \in L_i$ **do**
4     **if** $\mathtt{u}_j^i \leq 0$ **then**
5         create a node $v$ with state value $s(v) = s(u) + \eta_i(\mathtt{u}_j^i)$ (if it does not already exist) in the node layer $\mathcal{U}_{i+1}$
6     **else if** $\mathtt{l}_j^i \geq 0$ **then**
7         create a node $v$ with state value $s(v) = s(u) + \eta_i(\mathtt{l}_j^i)$ (if it does not already exist) in the node layer $\mathcal{U}_{i+1}$
8     **else**
9         create a node $v$ with state value $s(v) = s(u)$ (if it does not already exist) in the node layer $\mathcal{U}_{i+1}$
10     add two arcs from $u$ to $v$ with label values $\mathtt{l}_j^i$ and $\mathtt{u}_j^i$ respectively
11 **forall** $u \in \mathcal{U}_n$, $j \in L_n$ **do**
12     **if** $\mathtt{u}_j^i \leq 0$ **then**
13         calculate $\bar{s} = s(u) + \eta_i(\mathtt{u}_j^i)$
14     **else if** $\mathtt{l}_j^i \geq 0$ **then**
15         calculate $\bar{s} = s(u) + \eta_i(\mathtt{l}_j^i)$
16     **else**
17         calculate $\bar{s} = s(u)$
18     **if** $\bar{s} \leq \beta$ **then**
19         add two arcs from $u$ to the terminal node $t$ with label values $\mathtt{l}_j^i$ and $\mathtt{u}_j^i$ respectively

---

**Proposition 3.** *Consider $\mathcal{F} = \{cx \in \prod_{i=1}^{n}[\mathbf{l}^i, \mathbf{u}^i] \mid \eta(cx) \leq \beta\}$ where $n \in \mathbb{N}$. Let $\mathcal{D}$ be the DD constructed via Algorithm 1 for some domain partitioning sub-intervals $L_i$ for $i \in [n]$. Then, $\mathrm{conv}(\mathcal{F}) \subseteq \mathrm{conv}(\mathrm{Sol}(\mathcal{D}))$.*

*Proof.* It suffices to show that $\mathcal{F} \subseteq \mathrm{conv}(\mathrm{Sol}(\mathcal{D}))$ since the convex hull of a set is the smallest convex set that contains it. Pick $\bar{c}x \in \mathcal{F}$. It follows from the definition of $\mathcal{F}$ that $\sum_{i=1}^{n} \eta_i(\bar{x}_i) \leq \beta$. For each $i \in [n]$, let $j_i^*$ be the index of a domain sub-interval $[\mathbf{l}_{j_i^*}^i, \mathbf{u}_{j_i^*}^i]$ in $L_i$ such that $\mathbf{l}_{j_i^*}^i \leq \bar{x}_i \leq \mathbf{u}_{j_i^*}^i$. Such index exists because $\bar{x} \in [\mathbf{l}^i, \mathbf{u}^i] = \bigcup_{j \in L_i}[\mathbf{l}_j^i, \mathbf{u}_j^i]$ where the inclusion follows from the fact that $\bar{x} \in \mathcal{F}$, and the equality follows from the definition of domain partitioning. Next, we show that $\mathcal{D}$ includes a node sequence $u_1, u_2, \ldots, u_{n+1}$, where $u_i \in \mathcal{U}_i$ for $i \in [n+1]$, such that each node $u_i$ is connected to $u_{i+1}$ via two arcs with labels $\mathbf{l}_{j_i^*}^i$ and $\mathbf{u}_{j_i^*}^i$ for each $i \in [n]$. We prove the result using induction on the node layer index $k \in [n]$ in the node sequence $u_1, u_2, \ldots, u_{n+1}$. The induction base $k = 1$ follows from line 1 of Algorithm 1 as the root node $r$ can be considered as $u_1$. For the inductive hypothesis, assume that there exists a node sequence $u_1, u_2, \ldots, u_k$ of $\mathcal{D}$ with $u_j \in \mathcal{U}_j$ for $j \in [k]$ such that each node $u_i$ is connected to $u_{i+1}$ via two arcs with labels $\mathbf{l}_{j_i^*}^i$ and $\mathbf{u}_{j_i^*}^i$ for each $i \in [k-1]$. For the inductive step, we show that there exists a node sequence $u_1, u_2, \ldots, u_k, u_{k+1}$ of $\mathcal{D}$ with $u_j \in \mathcal{U}_j$ for $j \in [k+1]$ such that each node $u_i$ is connected to $u_{i+1}$ via two arcs with labels $\mathbf{l}_{j_i^*}^i$ and $\mathbf{u}_{j_i^*}^i$ for each $i \in [k]$. We consider two cases. For the first case, assume that $k \leq n-1$. Then, the for-loop lines 3–10 of Algorithm 1 imply that node $u_k$ is connected to another node in the node layer $\mathcal{U}_{k+1}$, which can be considered as $u_{k+1}$, via two arcs with labels $l_{j_k^*}^k$ and $u_{j_k^*}^k$ because the conditions of the for-loop are satisfied as follows: $k \in [n-1]$ due to the assumption of the first case, $u_k \in \mathcal{U}_k$ because of the inductive hypothesis, and $j_k^* \in L_k$ by construction. For the second case of the inductive step, assume that $k = n$. It follows from lines 1–10 of Algorithm 1 that the state value of node $u_{i+1}$ for $i \in [k-1]$ is calculated as $s(u_{i+1}) = s(u_i) + \gamma_i$ where $s(u_1) = 0$ because of line 1 of the algorithm, and where $\gamma_i$ is calculated depending on the sub-interval bounds $\mathbf{l}_{j_i^*}^i$ and $\mathbf{u}_{j_i^*}^i$ according to lines 4–9 of the algorithm. In particular, $\gamma_i = \eta_i(\mathbf{u}_{j_i^*}^i)$ if $\mathbf{u}_{j_i^*}^i \leq 0$, $\gamma_i = \eta_i(\mathbf{l}_{j_i^*}^i)$ if $\mathbf{l}_{j_i^*}^i \geq 0$, and $\gamma_i = 0$ otherwise. As a result, we have $s(u_k) = \sum_{i=1}^{k-1} \gamma_i$. On the other hand, since $\eta(cx)$ is a scale function, according to Definition 1, we must have $\eta_i(0) = 0$, $\eta_i(x_1) \leq \eta_i(x_2)$ for $0 \leq x_1 \leq x_2$, and $\eta_i(x_1) \geq \eta_i(x_2)$ for $x_1 \leq x_2 \leq 0$ for each $i \in [n]$. Using the fact that $\mathbf{l}_{j_i^*}^i \leq \bar{x}_i \leq \mathbf{u}_{j_i^*}^i$, we consider three cases. (i) If $\mathbf{u}_{j_i^*}^i \leq 0$, then $\bar{x}_i \leq \mathbf{u}_{j_i^*}^i \leq 0$, and thus $\eta_i(\bar{x}_i) \geq \eta_i(\mathbf{u}_{j_i^*}^i) = \gamma_i$. (ii) If $\mathbf{l}_{j_i^*}^i \geq 0$, then $\bar{x}_i \geq \mathbf{l}_{j_i^*}^i \geq 0$, and thus $\eta_i(\bar{x}_i) \geq \eta_i(\mathbf{l}_{j_i^*}^i) = \gamma_i$. (iii) If $\mathbf{u}_{j_i^*}^i > 0$ and $\mathbf{l}_{j_i^*}^i < 0$, then $\eta_i(\bar{x}_i) \geq 0 = \gamma_i$. Considering all these cases, we conclude that $\gamma_i \leq \eta_i(\bar{x}_i)$ for each $i \in [k-1]$. Therefore, we can write that $s(u_k) = \sum_{i=1}^{k-1} \gamma_i \leq \sum_{i=1}^{k-1} \eta_i(\bar{x}_i)$. Now consider lines 11-18 of the for-loop of Algorithm 1 for $u_k \in \mathcal{U}_k$ and $j_k^* \in L_k$. We compute $\bar{s} = s(u_k) + \gamma_k$, where $\gamma_k$ is calculated as described previously. Using a similar argument to that above, we conclude that $\gamma_k \leq \eta_k(\bar{x}_k)$. Combining this result with that derived for $s(u_k)$, we obtain that $\bar{s} = \sum_{i=1}^{k} \gamma_i \leq \sum_{i=1}^{k} \eta_i(\bar{x}_i) \leq \beta$, where the last inequality follows from the fact that $\bar{c}x \in \mathcal{F}$. Therefore, lines 18–19 of Algorithm 1 imply that two arcs with label values $l_{j_k^*}^k$ and $u_{j_k^*}^k$ connect node $u_k$ to the terminal node $t$ which can be considered as $u_{k+1}$, completing the desired node sequence. Now consider the collection of points $\tilde{c}x^\kappa$ for $\kappa \in [2^n]$ encoded by all paths composed of the above-mentioned pair of arcs with labels $\mathbf{l}_{j_i^*}^i$ and $\mathbf{u}_{j_i^*}^i$ between each two consecutive nodes $u_i$ and $u_{i+1}$ in the sequence $u_1, u_2, \ldots, u_{n+1}$. Therefore, $\tilde{c}x^\kappa \in \mathrm{Sol}(\mathcal{D})$ for $\kappa \in [2^n]$. It is clear that these points also form the vertices of an $n$-dimensional hyper-rectangle defined by $\prod_{i=1}^{n}[\mathbf{l}_{j_i^*}^i, \mathbf{u}_{j_i^*}^i]$. By construction, we have that $\bar{c}x \in \prod_{i=1}^{n}[\mathbf{l}_{j_i^*}^i, \mathbf{u}_{j_i^*}^i]$, i.e., $\bar{c}x$ is a point inside the above hyper-rectangle. As a result, $\bar{c}x$ can be represented as a convex combination of the vertices $\tilde{c}x^\kappa$ for $\kappa \in [2^n]$ of the hyper-rectangle, yielding $\bar{c}x \in \mathrm{conv}(\mathrm{Sol}(\mathcal{D}))$. $\qquad\square$

In view of Proposition 3, note that the variable domains are not required to be bounded, as the state value calculated for each node of the DD is always finite. This allows for the construction of the entire DD layers regardless of the specific values of the arc labels. This property is specifically useful when DDs are employed to build convex relaxations for problems that do not have initial finite bounds on variables, such as statistical estimation models where the parameters can take any value in $\mathbb{R}$.

**Remark 1.** *The size of the DD obtained from Algorithm 1 could grow exponentially when the number of variables and sub-intervals in the domain partitions increases. To control this growth rate, there are two common approaches. The first approach involves controlling the size of the DD by reducing the number of sub-intervals in the domain partitions of variables at certain layers. For example, assume that the number of nodes at each layer of the DD grows through the top-down construction procedure of Algorithm 1 until it reaches the imposed width limit $W$ at layer $k$. Then, by setting the number of sub-intervals for the next layer $k + 1$ to one (i.e., $|L_{k+1}| = 1$), the algorithm guarantees that the number of nodes at layer $k + 1$ will not exceed $W$. The second approach involves controlling the size of the DD by creating a "relaxed DD" through merging nodes at layers that exceed the width limit $W$. In this process, multiple nodes $\{v_1, v_2, \ldots, v_k\}$, for some $k \in \mathbb{N}$, in a layer are merged into a single node $\tilde{v}$ in such a way that all feasible paths of the original DD are maintained. For the DD constructed through Algorithm 1, choosing the state value $s(\tilde{v}) = \min_{j=1,\ldots,k}\{s(v_j)\}$ provides such a guarantee for all feasible paths of the original DD to be maintained, because this state value underestimates the state values of each of the merged nodes, producing a relaxation for the norm-bounding constraint $\eta(\boldsymbol{c}x) \leq \beta$; see Appendix 2 and references therein for an exposition to DD relaxations.*

### 4.2 OUTER APPROXIMATION

The next step after constructing a DD that provides a relaxation for the convex hull of a set is to obtain an explicit description of the convex hull of the DD solution set to be implemented inside an outer approximation framework to find dual bounds. Recently, Davarnia (2021); Davarnia & Van Hoeve (2020) proposed efficient methods to obtain such convex hull descriptions for DDs in the original space of variables through a successive generation of cutting planes. In Appendix A.1, we present a summary of those methods that are applicable for the DD built by Algorithm 1.

## 5 SPATIAL BRANCH-AND-CUT

In global optimization of MINLPs, a divide-and-conquer strategy, such as spatial branch-and-bound, is employed to achieve convergence to the global optimal value of the problem. The spatial branch-and-bound strategy reduces the variables' domain through successive partitioning of the original box domains of the variables. These partitions are often rectangular in shape, dividing the variables' domain into smaller hyper-rectangles as a result of branching. For each such partition, a convex relaxation is constructed to calculate a dual bound. Throughout this process, the dual bounds are updated as tighter relaxations are obtained, until they converge to a specified vicinity of the global optimal value of the problem. To prove such converge results, one needs to show that the convexification method employed at each domain partition converges (in the Hausdorff sense) to the convex hull of the feasible region restricted to that partition; see Belotti et al. (2009); Ryoo & Sahinidis (1996); Tawarmalani & Sahinidis (2004) for a detailed account of spatial branch-and-bound methods for MINLPs.

As demonstrated in the previous section, the convexification method used in our framework involves generating cutting planes for the solution set of the DDs obtained from Algorithm 1. We refer to the spatial branch-and-bound strategy incorporated into our solution method as *spatial branch-and-cut* (SB&C). In this section, we show that the convex hull of the solution set of the DDs obtained from Algorithm 1 converges to the convex hull of the solutions of the original set $\mathcal{F}$ as the partition volume reduces. First, we prove that reducing the variables' domain through partitioning produces tighter convex relaxations obtained by the proposed DD-based convexification method described in Section 4.

**Proposition 4.** *Consider $\mathcal{F}_j = \{\boldsymbol{c}x \in P_j \,|\, \eta(\boldsymbol{c}x) \leq \beta\}$ for $j = 1, 2$, where $\eta(\boldsymbol{c}x)$ is a scale function, and $P_j = \prod_{i=1}^{n}[\mathtt{l}_j^i, \mathtt{u}_j^i]$ is a box domain of variables. For each $j = 1, 2$, let $\mathcal{D}_j$ be the DD constructed via Algorithm 1 for a single sub-interval $[\mathtt{l}_j^i, \mathtt{u}_j^i]$ for $i \in [n]$. If $P_2 \subseteq P_1$, then $\mathrm{conv}(\mathrm{Sol}(\mathcal{D}_2)) \subseteq \mathrm{conv}(\mathrm{Sol}(\mathcal{D}_1))$.*

*Proof.* Since there is only one sub-interval $[\mathtt{l}_2^i, \mathtt{u}_2^i]$ for the domain of variable $x_i$ for $i \in [n]$, there is only one node, referred to as $u_i$, at each node layer of $\mathcal{D}_2$ according to Algorithm 1. Following the top-down construction steps of this algorithm, for each $i \in [n-1]$, $u_i$ is connected via two arcs with label values $\mathtt{l}_2^i$ and $\mathtt{u}_2^i$ to $u_{i+1}$. There are two cases for the arcs at layer $i = n$ based on the

value of $\bar{s}$ calculated in lines 12–17 of Algorithm 1. For the first case, assume that $\bar{s} > \beta$. Then, the if-condition in line 18 of Algorithm 1 is not satisfied. Therefore, node $u_n$ is not connected to the terminal node $t$ of $\mathcal{D}_2$. As a result, there is no $r$-$t$ path in this DD, leading to an empty solution set, i.e., $\mathrm{conv}(\mathrm{Sol}(\mathcal{D}_2)) = \mathrm{Sol}(\mathcal{D}_2) = \emptyset$. This proves the result since $\emptyset \subseteq \mathrm{conv}(\mathrm{Sol}(\mathcal{D}_1))$. For the second case, assume that $\bar{s} \leq \beta$. Then, the if-condition in line 18 of Algorithm 1 is satisfied, and node $u_n$ is connected to the terminal node $t$ of $\mathcal{D}_2$ via two arcs with label values $\mathtt{l}_2^n$ and $\mathtt{u}_2^n$. Therefore, the solution set of $\mathcal{D}_2$ contains $2^n$ points encoded by all the $r$-$t$ paths of the DD, each composed of arcs with label values $\mathtt{l}_2^i$ or $\mathtt{u}_2^i$ for $i \in [n]$. It is clear that these points correspond to the extreme points of the rectangular partition $P_2 = \prod_{i=1}^{n}[\mathtt{l}_2^i, \mathtt{u}_2^i]$. Pick one of these points, denoted by $\bar{c}x$. We show that $\bar{c}x \in \mathrm{conv}(\mathrm{Sol}(\mathcal{D}_1))$. It follows from lines 1–10 of Algorithm 1 that each layer $i \in [n]$ includes a single node $v_i$. Further, each node $v_i$ is connected to $v_{i+1}$ via two arcs with label values $\mathtt{l}_1^i$ and $\mathtt{u}_1^i$ for $i \in [n-1]$. To determine whether $v_n$ is connected to the terminal node of $\mathcal{D}_1$, we need to calculate $\bar{s}$ (which we refer to as $\bar{\bar{s}}$ to distinguish it from that calculated for $\mathcal{D}_2$) according to lines 12–17 of Algorithm 1. Using an argument similar to that in the proof of Proposition 3, we write that $\bar{\bar{s}} = \sum_{i=1}^{n} s(v_i)$, where $s(v_i) = \eta_i(\mathtt{u}_1^i)$ if $\mathtt{u}_1^i \leq 0$, $s(v_i) = \eta_i(\mathtt{l}_1^i)$ if $\mathtt{l}_1^i \geq 0$, and $s(v_i) = 0$ otherwise for all $i \in [n]$, and $s(v_1) = 0$. On the other hand, we can similarly calculate the value of $\bar{s}$ for $\mathcal{D}_2$ as $\bar{s} = \sum_{i=1}^{n} s(u_i)$, where $s(u_i) = \eta_i(\mathtt{u}_2^i)$ if $\mathtt{u}_2^i \leq 0$, $s(u_i) = \eta_i(\mathtt{l}_2^i)$ if $\mathtt{l}_2^i \geq 0$, and $s(u_i) = 0$ otherwise for all $i \in [n]$, and $s(u_1) = 0$. Because $P_2 \subseteq P_1$, we have that $\mathtt{l}_1^i \leq \mathtt{l}_2^i \leq \mathtt{u}_2^i \leq \mathtt{u}_1^i$ for each $i \in [n]$. Consider three cases. If $\mathtt{u}_2^i \leq 0$, then either $\mathtt{u}_1^i \leq 0$ which leads to $s(v_i) = \eta_i(\mathtt{u}_1^i) \leq \eta_i(\mathtt{u}_2^i) = s(u_i)$ due to monotone property of scale functions, or $\mathtt{u}_1^i > 0$ which leads to $s(v_i) = 0 \leq \eta_i(\mathtt{u}_2^i) = s(u_i)$ as in this case $\mathtt{l}_1^i \leq \mathtt{l}_2^i \leq \mathtt{u}_2^i \leq 0$. If $\mathtt{l}_2^i \geq 0$, then either $\mathtt{l}_1^i \geq 0$ which leads to $s(v_i) = \eta_i(\mathtt{l}_1^i) \leq \eta_i(\mathtt{l}_2^i) = s(u_i)$ due to monotone property of scale functions, or $\mathtt{l}_1^i < 0$ which leads to $s(v_i) = 0 \leq \eta_i(\mathtt{l}_2^i) = s(u_i)$ as in this case $\mathtt{u}_1^i \geq \mathtt{u}_2^i \geq \mathtt{l}_2^i \geq 0$. If $\mathtt{u}_2^i > 0$ and $\mathtt{l}_2^i < 0$, then $s(v_i) = s(u_i) = 0$. As a result, $s(v_i) \leq s(u_i)$ for all cases and for all $i \in [n]$. Therefore, we obtain that $\bar{\bar{s}} = \sum_{i=1}^{n} s(v_i) \leq \sum_{i=1}^{n} s(u_i) = \bar{s} \leq \beta$, where the last inequality follows from the assumption of this case. We conclude that the if-condition in line 18 of Algorithm 1 is satisfied for $\mathcal{D}_1$, and thus $v_n$ is connected to the terminal node of $\mathcal{D}_2$ via two arcs with label values $\mathtt{l}_1^n$ and $\mathtt{u}_1^n$. Consequently, $\mathrm{Sol}(\mathcal{D}_1)$ includes all extreme points of the rectangular partition $P_1$ encoded by the $r$-$t$ paths of this DD. Since $P_2 \subseteq P_1$, the extreme point $\bar{c}x$ of $P_2$ is in $\mathrm{conv}(\mathrm{Sol}(\mathcal{D}_1))$, proving the result. $\qquad \square$

While Proposition 4 implies that the dual bounds obtained by the DD-based convexification framework can improve through SB&C as a result of partitioning the variables' domain, additional functional properties for the scale function in the norm-bounding constraint are required to ensure convergence to the global optimal value of the problem. Next, we show that a sufficient condition to guarantee such convergence is the lower semicontinuity of the scale function. This condition is not very restrictive, as all $\ell_p$-norm functions for $p \in [0, \infty)$ as well as typical nonconvex penalty functions, such as those described in Proposition 1, satisfy this condition.

**Proposition 5.** *Consider a scale function $\eta(\boldsymbol{c}x) = \sum_{i=1}^{n} \eta_i(x_i)$, where $\eta_i(x) : \mathbb{R} \to \mathbb{R}_+$ is lower semicontinuous for $i \in [n]$, i.e., $\liminf_{x \to x_0} \eta_i(x) \geq \eta_i(x_0)$ for all $x_0 \in \mathbb{R}$. Define $\mathcal{F}_j = \{\boldsymbol{c}x \in P_j \mid \eta(\boldsymbol{c}x) \leq \beta\}$ for $j \in \mathbb{N}$, where $P_j = \prod_{i=1}^{n}[\mathtt{l}_j^i, \mathtt{u}_j^i]$ is a bounded box domain of variables. For $j \in \mathbb{N}$, let $\mathcal{D}_j$ be the DD associated with $\mathcal{F}_j$ that is constructed via Algorithm 1 for single sub-interval $[\mathtt{l}_j^i, \mathtt{u}_j^i]$ for $i \in [n]$. Assume that $\{P_1, P_2, \dots\}$ is a monotone decreasing sequence of rectangular partitions of the variables domain created through the SB&C process, i.e., $P_j \supset P_{j+1}$ for each $j \in \mathbb{N}$. Let $\tilde{c}x \in \mathbb{R}^n$ be the point in a singleton set to which the above sequence converges (in the Hausdorff sense), i.e., $\{P_j\} \downarrow \{\tilde{c}x\}$. Then, the following statements hold:*

*(i) If $\eta(\tilde{c}x) \leq \beta$, then $\big\{ \mathrm{conv}(\mathrm{Sol}(\mathcal{D}_j)) \big\} \downarrow \{\tilde{c}x\}$.*

*(ii) If $\eta(\tilde{c}x) > \beta$, then there exists $m \in \mathbb{N}$ such that $\mathrm{Sol}(\mathcal{D}_j) = \emptyset$ for all $j \geq m$.*

*Proof.* (i) Assume that $\eta(\tilde{c}x) \leq \beta$. Consider $j \in \mathbb{N}$. First, note that $\mathcal{F}_j \subseteq \mathrm{conv}(\mathcal{F}_j) \subseteq \mathrm{conv}(\mathrm{Sol}(\mathcal{D}_j))$ according to Proposition 3. Next, we argue that $\mathrm{Sol}(\mathcal{D}_j) \subseteq P_j$. There are two cases. For the first case, assume that the if-condition in line 18 of Algorithm 1 is violated. Then, it implies that there are no $r$-$t$ paths in $\mathcal{D}_j$, i.e., $\mathrm{Sol}(\mathcal{D}_j) = \emptyset \subseteq P_j$. For the second case, assume that the if-condition in line 18 of Algorithm 1 is satisfied. Then, it implies that $\mathrm{Sol}(\mathcal{D}_j)$ contains the points encoded by all $r$-$t$ paths in $\mathcal{D}_j$ composed of arc label values $\mathtt{l}_j^i$ or $\mathtt{u}_j^i$ for each $i \in [n]$, i.e., $\mathrm{Sol}(\mathcal{D}_j) \subseteq P_j$. As a result, $\mathrm{conv}(\mathrm{Sol}(\mathcal{D}_j)) \subseteq$

$P_j$. It follows from Proposition 4 that the sequence $\{\mathrm{conv}(\mathrm{Sol}(\mathcal{D}_1)), \mathrm{conv}(\mathrm{Sol}(\mathcal{D}_2)), \dots\}$ is monotone non-increasing, i.e., $\mathrm{conv}(\mathrm{Sol}(\mathcal{D}_j)) \supseteq \mathrm{conv}(\mathrm{Sol}(\mathcal{D}_{j+1}))$ for $j \in \mathbb{N}$. On the other hand, we can write $\mathcal{F}_j = \{cx \in \mathbb{R}^n \,|\, \eta(cx) \le \beta\} \cap P_j$ by definition. Since $\{P_j\} \downarrow \{\tilde{c}x\}$, we obtain that $\{\mathcal{F}_j\} \downarrow \{cx \in \mathbb{R}^n \,|\, \eta(cx) \le \beta\} \cap \{\tilde{c}x\} = \{\tilde{c}x\}$ since $\eta(\tilde{c}x) \le \beta$ by assumption. Therefore, based on the previous arguments, we can write that $\mathcal{F}_j \subseteq \mathrm{conv}(\mathrm{Sol}(\mathcal{D}_j)) \subseteq P_j$. Because $\{\mathcal{F}_j\} \downarrow \{\tilde{c}x\}$ and $\{P_j\} \downarrow \{\tilde{c}x\}$, we conclude that $\big\{ \mathrm{conv}(\mathrm{Sol}(\mathcal{D}_j)) \big\} \downarrow \{\tilde{c}x\}$.

(ii) Assume that $\eta(\tilde{c}x) > \beta$. Since the sequence of domain partitions $\{P_j\}$ is a monotone decreasing sequence that converges to $\{\tilde{c}x\}$, we can equivalently write that the sequence of variable lower bounds $\{\mathtt{l}_1^i, \mathtt{l}_2^i, \dots\}$ in these partitions is monotone non-decreasing and it converges to $\tilde{x}_i$ for $i \in [n]$, i.e., $\mathtt{l}_1^i \le \mathtt{l}_2^i \le \dots$, and $\lim_{j \to \infty} \mathtt{l}_j^i = \tilde{x}_i$. Similarly, the sequence of variable upper bounds $\{\mathtt{u}_1^i, \mathtt{u}_2^i, \dots\}$ in these partitions is monotone non-increasing and it converge to $\tilde{x}_i$ for $i \in [n]$, i.e., $\mathtt{u}_1^i \ge \mathtt{u}_2^i \ge \dots$, and $\lim_{j \to \infty} \mathtt{u}_j^i = \tilde{x}_i$. The assumption of this case implies that $\eta(\tilde{c}x) = \sum_{i=1}^n \eta_i(\tilde{x}_i) > \beta$. Define $\epsilon = \frac{\sum_{i=1}^n \eta_i(\tilde{x}_i) - \beta}{n} > 0$. For each DD $\mathcal{D}_j$ for $j \in \mathbb{N}$, using an argument similar to that of Proposition 4, we can calculate the value $\bar{s}$ in line 11–17 of Algorithm 1 as $\bar{s} = \sum_{i=1}^n s(v_i)$ where $v_i$ is the single node at layer $i \in [n]$ of $\mathcal{D}_j$. In this equation, considering that the variable domain is $[\mathtt{l}_j^i, \mathtt{u}_j^i]$ for each $i \in [n]$, we have $s(v_1) = 0$, and $s(v_i) = \eta_i(\mathtt{u}_j^i)$ if $\mathtt{u}_j^i \le 0$, $s(v_i) = \eta_i(\mathtt{l}_j^i)$ if $\mathtt{l}_j^i \ge 0$, and $s(v_i) = 0$ otherwise. Because $\eta_i(x_i)$, for $i \in [n]$, is lower semicontinuous at $\tilde{x}_i$ by assumption, for every real $z_i < \eta_i(\tilde{x}_i)$, there exists a nonempty neighborhood $\mathcal{R}_i = (\hat{\mathtt{l}}^i, \hat{\mathtt{u}}^i)$ of $\tilde{x}_i$ such that $\eta_i(x) > z_i$ for all $x \in \mathcal{R}_i$. Set $z_i = \eta_i(\tilde{x}_i) - \epsilon$. On the other hand, since $\lim_{j \to \infty} \mathtt{l}_j^i = \tilde{x}_i$ and $\lim_{j \to \infty} \mathtt{u}_j^i = \tilde{x}_i$, by definition of sequence convergence, there exists $m_i \in \mathbb{N}$ such that $\mathtt{l}_{m_i}^i > \hat{\mathtt{l}}^i$ and $\mathtt{u}_{m_i}^i < \hat{\mathtt{u}}^i$. Pick $m = \max_{i \in [n]} m_i$. It follows that, for each $i \in [n]$, we have $\eta_i(x) > z_i = \eta_i(\tilde{x}_i) - \epsilon$ for all $x \in [\mathtt{l}_m^i, \mathtt{u}_m^i]$. Now, consider the value of $s(v_i)$ at layer $i \in [n]$ of $\mathcal{D}_m$. There are three cases. If $\mathtt{u}_m^i \le 0$, then $s(v_i) = \eta_i(\mathtt{u}_m^i) > \eta_i(\tilde{x}_i) - \epsilon$. If $\mathtt{l}_m^i \ge 0$, then $s(v_i) = \eta_i(\mathtt{l}_m^i) > \eta_i(\tilde{x}_i) - \epsilon$. If $\mathtt{u}_m^i > 0$ and $\mathtt{l}_m^i < 0$, then $s(v_i) = \eta_i(0) = 0 > \eta_i(\tilde{x}_i) - \epsilon$ as $0 \in [\mathtt{l}_m^i, \mathtt{u}_m^i]$. Therefore, $s(v_i) > \eta_i(\tilde{x}_i) - \epsilon$ for all cases. Using the arguments given previously, we calculate that $\bar{s} = \sum_{i=1}^n s(v_i) > \sum_{i=1}^n \eta_i(\tilde{x}_i) - n\epsilon = \beta$, where the last equality follows from the definition of $\epsilon$. Since $\bar{s} > \beta$, the if-condition in line 18 of Algorithm 1 is not satisfied, and thus node $v_n$ is not connected to the terminal node in DD $\mathcal{D}_m$, implying that $\mathrm{Sol}(\mathcal{D}_m) = \emptyset$. Finally, it follows from Proposition 4 that $\mathrm{Sol}(\mathcal{D}_j) \subseteq \mathrm{conv}(\mathrm{Sol}(\mathcal{D}_j)) \subseteq \mathrm{conv}(\mathrm{Sol}(\mathcal{D}_m)) = \emptyset$, for all $j > m$, proving the result.

$\square$

The result of Proposition 5 shows that the convex hull of the solution set represented by the DDs constructed through our convexification technique converges to the feasible region of the underlying norm-bounding constraint during the SB&C process. If this constraint is embedded into an optimization problem whose other constraints also have a convexification method with such convergence results, it can guarantee convergence to the global optimal value of the optimization problem through the incorporation of SB&C.

We conclude this section with two remarks about the results of Proposition 5. First, although the above convergence results are obtained for DDs with a unit width $W = 1$, they can be easily extended to cases with larger widths using the following observation. The solution set of a DD can be represented by a finite union of the solution sets of DDs of width 1 obtained from decomposing node sequences of the original DD. By considering the collection of all such decomposed DDs of unit width, we can use the result of Proposition 5 to show convergence to the feasible region of the underlying constraint. In practice, increasing the width limit of a DD often leads to stronger DDs with tighter convex relaxations. This, in turn, accelerates the bound improvement rate, helping to achieve the desired convergence results faster.

Second, even though the convergence results of Proposition 5 are proven for a bounded domain of variables, the SB&C can still be implemented for bounded optimization problems that contain unbounded variables. In this case, as is common in spatial branch-and-bound solution methods, the role of a primal bound becomes critical for pruning nodes that contain very large (or infinity) variable bounds, which result in large dual bounds. This is particularly evident in the statistical models that

minimize an objective function composed of the estimation errors and a penalty function based on parameters' norms. In the next section, we provide preliminary computational results for an instance of such models.

## 6 COMPUTATIONAL RESULTS

In this section, we present preliminary computational results to evaluate the effectiveness of our solution framework. As previously outlined, although our framework can be applied to a broad class of norm-type constraints with general MINLP structures, here we focus on a well-known model that involves a challenging nonconvex norm-induced penalty function. In particular, we consider the following statistical regression problem with the SCAD penalty function.

$$\min_{\boldsymbol{c}\beta\in\mathbb{R}^p} \quad ||\boldsymbol{c}y - X\boldsymbol{c}\beta||_2^2 + \rho^{\texttt{SCAD}}(\boldsymbol{c}\beta, \boldsymbol{c}\lambda, \boldsymbol{c}\gamma), \tag{1}$$

where $n$ is the sample size, $p$ is the number of features, $X \in \mathbb{R}^{n\times p}$ is the data matrix, $\boldsymbol{c}y \in \mathbb{R}^n$ is the response vector, $\boldsymbol{c}\beta \in \mathbb{R}^n$ is the decision variables that represent coefficient parameters in the regression model, and $\rho^{\texttt{SCAD}}(\boldsymbol{c}\beta, \boldsymbol{c}\lambda, \boldsymbol{c}\gamma)$ is the SCAD penalty function as described in Proposition 1. It follows from this proposition that $\rho^{\texttt{SCAD}}(\boldsymbol{c}\beta, \boldsymbol{c}\lambda, \boldsymbol{c}\gamma)$ is a scale function, making the problem amenable to our DD-based solution framework.

We note that the SCAD penalty function is chosen as a challenging nonconvex regularization structure to showcase the capabilities of our framework in handling such structures. This is an important test feature because the SCAD penalty function is not admissible in state-of-the-art global solvers, such as BARON, due to the presence of the integration operator. As a result, to our knowledge, our proposed approach is the first general-purpose global solution framework for this problem class.

For our experiments, we use datasets from public repositories such as UCI Machine Learning Repository Kelly et al. and Kaggle Kag. The collection of these datasets, described by their size pair $(p, n)$, is $\{(7, 400), (36, 4434), (60, 207), (127, 123), (147, 168), (384, 4000)\}$. More details about these instances and the settings of our DD-based algorithm can be found in Appendix A.2. Here, we present a summary of the results of our global solution framework in Figure 1. These figures show the solution time to achieve an optimality gap of $< 5\%$ on the vertical axis and the instance number in the above collection on the horizontal axis; see Appendix A.2 for detailed numerical results. We have conducted these experiments for two different categories of $(\lambda, \gamma) \in \{(1, 3), (10, 30)\}$ to consider both small and large degrees of regularization and nonconvdxity for the SCAD function. These results demonstrate the potential of our solution method in globally solving equation 1 for small to medium-sized problem instances.

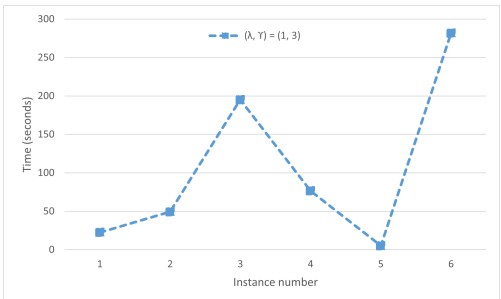 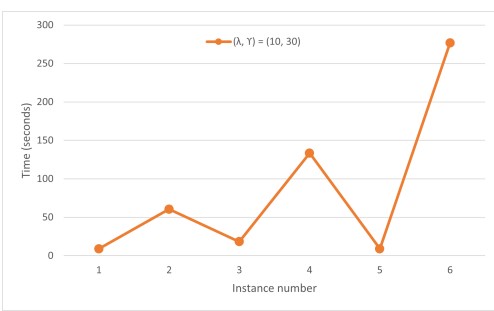

Figure 1: Solution time of the DD framework for selected datasets with different penalty parameters

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

# A  APPENDIX

## A.1  CONVEX HULL DERIVATION

In this section, we present a summary of the results in Davarnia (2021); Davarnia & Van Hoeve (2020) that can be used to obtain a convex hull description for the DDs constructed in Section 4.1. We start with the description of the convex hull in an extended (lifted) space of variables.

**Proposition 6.** *Consider a DD $\mathcal{D} = (\mathcal{U}, \mathcal{A}, l(.))$ with solution set $\mathrm{Sol}(\mathcal{D}) \subseteq \mathbb{R}^n$ that is obtained from Algorithm 1 associated with the constraint $\mathcal{F} = \{cx \in \prod_{i=1}^{n} [\mathtt{l}^i, \mathtt{u}^i] \,|\, \eta(cx) \leq \beta\}$. Define $\mathcal{G}(\mathcal{D}) = \{(cx; cy) \in \mathbb{R}^n \times \mathbb{R}^{|\mathcal{A}|} \,|\, \text{equation } 2a, \text{equation } 2b\}$ where*

$$\sum_{a \in \delta^+(u)} y_a - \sum_{a \in \delta^-(u)} y_a = f_u, \qquad \forall u \in \mathcal{U} \tag{2a}$$

$$\sum_{a \in \mathcal{A}_k} l(a)\, y_a = x_k, \qquad \forall k \in [n] \tag{2b}$$

$$y_a \geq 0, \qquad \forall u \in \mathcal{U}, \tag{2c}$$

*where $f_s = -f_t = 1$, $f_u = 0$ for $u \in \mathcal{U} \setminus \{s, t\}$, and $\delta^+(u)$ (resp. $\delta^-(u)$) denotes the set of outgoing (resp. incoming) arcs at node $u$. Then, $\mathrm{proj}_x \mathcal{G}(\mathcal{D}) = \mathrm{conv}(\mathrm{Sol}(\mathcal{D}))$.* □

Viewing $y_a$ as the network flow variable on arc $a \in \mathcal{A}$ of $\mathcal{D}$, the formulation equation 2a–equation 2a implies that the LP relaxation of the network model that routes one unit of supply from the root node to the terminal node of the DD represents a convex hull description for the solution set of $\mathcal{D}$ in a higher dimension. Thus, projecting the arc-flow variables $cy$ from this formulation would yield $\mathrm{conv}(\mathrm{Sol}(\mathcal{D}))$ in the original space of variables. This result leads to a separation oracle that can be used to separate any point $\bar{c}x \in \mathbb{R}^n$ from $\mathrm{conv}(\mathrm{Sol}(\mathcal{D}))$ through solving the following LP.

**Proposition 7.** *Consider a DD $\mathcal{D} = (\mathcal{U}, \mathcal{A}, l(.))$ with solution set $\mathrm{Sol}(\mathcal{D}) \subseteq \mathbb{R}^n$ that is obtained from Algorithm 1 associated with the constraint $\mathcal{F} = \{cx \in \prod_{i=1}^{n} [\mathtt{l}^i, \mathtt{u}^i] \,|\, \eta(cx) \leq \beta\}$. Consider a point $\bar{c}x \in \mathbb{R}^n$, and define*

$$\omega^* = \max \quad \sum_{k \in [n]} \bar{x}_k \gamma_k - \theta_t \tag{3}$$

$$\theta_{\mathtt{t}(a)} - \theta_{\mathtt{h}(a)} + l(a)\gamma_k \leq 0, \qquad \forall k \in [n], a \in \mathcal{A}_k \tag{4}$$

$$\theta_s = 0, \tag{5}$$

*where $\mathtt{t}(a)$ and $\mathtt{h}(a)$ represent the tail and the head node of arc $a$, respectively. Then, $\bar{c}x \in \mathrm{conv}(\mathrm{Sol}(\mathcal{D}))$ if $\omega^* = 0$. Otherwise, $\bar{c}x$ can be separated from $\mathrm{conv}(\mathrm{Sol}(\mathcal{D}))$ via $\sum_{k \in [n]} x_k \gamma_k^* \leq \theta_t^*$ where $(c\theta^*; c\gamma^*)$ is an optimal recession ray of equation 3–equation 5.* □

The above separation oracle requires solving a LP whose size is proportional to the number of nodes and arcs of the DD, which could be computationally demanding when used repeatedly inside an outer approximation framework. As a result, an alternative subgradient-type method is proposed to solve the same separation problem, but with a focus on detecting a violated cut faster.

To summarize the recursive step of the separation method in Algorithm 2, the vector $c\gamma^{(\tau)} \in \mathbb{R}^n$ is used in line 3 to assign weights to the arcs of the DD, in which a longest $r$-$t$ path is obtained. The solution $cx^{\mathrm{P}^{(\tau)}}$ corresponding to this longest path is then subtracted from the separation point $\bar{c}x$, which provides the subgradient value for the objective function of the separation problem at point $c\gamma^{(\tau)}$. Then, the subgradient direction is updated in line 7 for a step size $\rho_\tau$, which is then projected onto the unit sphere on variables $c\gamma$ in line 8. It is shown that for an appropriate step size, this algorithm converges to an optimal solution of the separation problem equation 3–equation 5, which yields the desired cutting plane in line 11. Since this algorithm is derivative-free, as it calculates the subgradient values through solving a longest path problem over a weighted DD, it is very efficient in finding a violated cutting plane in comparison to the LP equation 3–equation 5, which makes it suitable for implementation inside spatial branch-and-cut frameworks such as that proposed in Section 5.

The cutting planes obtained from the separation oracles in Proposition 7 and Algorithm 2 can be employed inside an outer approximation framework as follows. We solve a convex relaxation of

---

**Algorithm 2:** A subgradient-based separation algorithm

---

**Data:** A DD $\mathcal{D} = (\mathcal{U}, \mathcal{A}, l(.))$ representing $\mathcal{F} = \{\boldsymbol{c}x \in \prod_{i=1}^{n}[\mathtt{1}^i, \mathtt{u}^i] \mid \eta(\boldsymbol{c}x) \leq \beta\}$ and a point $\bar{\boldsymbol{c}}x$

**Result:** A valid inequality to separate $\bar{\boldsymbol{c}}x$ from $\mathrm{conv}(\mathrm{Sol}(\mathcal{D}))$

1 initialize $\tau = 0$, $\boldsymbol{c}\gamma^{(0)} \in \mathbb{R}^n$, $\tau^* = 0$, $\Delta^* = 0$

2 **while** *the stopping criterion is not met* **do**

3      assing weights $w_a = l(a)\gamma_k^{(\tau)}$ to each arc $a \in \mathcal{A}_k$ of $\mathcal{D}$ for all $k \in [n]$

4      find a longest $r$-$t$ path $\mathrm{P}^{(\tau)}$ in the weighted DD and compute its encoding point $\boldsymbol{c}x^{\mathrm{P}^{(\tau)}}$

5      **if** $\boldsymbol{c}\gamma^{(\tau)}(\bar{\boldsymbol{c}}x - \boldsymbol{c}x^{\mathrm{P}^{(\tau)}}) > \max\{0, \Delta^*\}$ **then**

6         update $\tau^* = \tau$ and $\Delta^* = \boldsymbol{c}\gamma^{(\tau)}(\bar{\boldsymbol{c}}x - \boldsymbol{c}x^{\mathrm{P}^{(\tau)}})$

7      update $\boldsymbol{c}\phi^{(\tau+1)} = \boldsymbol{c}\gamma^{(\tau)} + \rho_\tau(\bar{\boldsymbol{c}}x - \boldsymbol{c}x^{\mathrm{P}^{(\tau)}})$ for step size $\rho_\tau$

8      find the projection $\boldsymbol{c}\gamma^{(\tau+1)}$ of $\boldsymbol{c}\phi^{(\tau+1)}$ onto the unit sphere defined by $||\boldsymbol{c}\gamma||_2 \leq 1$

9      set $\tau = \tau + 1$

10 **if** $\Delta^* > 0$ **then**

11      return inequality $\boldsymbol{c}\gamma^{(\tau^*)}(\boldsymbol{c}x - \boldsymbol{c}x^{\mathrm{P}^{(\tau^*)}}) \leq 0$

---

the problem whose optimal solution is denoted by $\boldsymbol{c}x^*$. Then, this solution is evaluated at $\mathcal{F}$. If $\eta(\boldsymbol{c}x^*) \leq \beta$, then the algorithm terminates due to finding a feasible (or optimal) solution of the problem. Otherwise, the above separation oracles are invoked to generate a cutting plane that separates $\boldsymbol{c}x^*$ from $\mathrm{conv}(\mathrm{Sol}(\mathcal{D}))$. The resulting cutting plane is added to the problem relaxation, and the procedure is repeated until no new cuts are added or a stopping criterion, such as iteration number or gap tolerance, is triggered. If at termination, an optimal solution is not returned, a spatial branch-and-bound scheme is employed as discussed in Section 5.

## A.2 Supplemental Computational Results

In this section, we give a detailed discussion about algorithmic settings and numerical results presented in Section 6. These results are obtained on a Windows 11 (64-bit) operating system, 64 GB RAM, 3.8 GHz AMD Ryzen CPU. The DD-ECP Algorithm is written in Julia v1.9 via JuMP v1.11.1, and the outer approximation models are solved with CPLEX v22.1.0.

For the DD construction, we use Algorithm 1 together with the DD relaxation technique described in Remark 1 with width limit $W$ set to 10000. The merging operation merges nodes with close state values that lie in the same interval of the state range. We set the number of sub-intervals $|L_i|$ up to 2500 for each DD layer $i$. To generate cutting planes for the outer approximation approach, we use the subgradient method of Algorithm 2 with the stopping criterion equal to 50 iterations. The maximum number of iterations for applying the subgradient method before invoking the branching procedure to create new children nodes is 100. The branching process selects the variable whose optimal value in the outer approximation model of the current node is closest to the middle point of its domain interval, and then the branching occurs at that value. Throughout the process, a primal bound is updated by constructing a feasible solution of the problem through plugging the current optimal solution of the outer approximation at each node into the objective function of equation 1. This primal bound is used to prune nodes with worse dual bounds than the current primal bound. The stopping criterion for our algorithm is reaching a relative optimality gap of $5\%$ which is calculated as (primal bound $-$ dual bound) / dual bound.

The numerical results obtained for instances we studied are presented in Tables 1 and 2. The first and second columns of these tables include the name of the dataset and the resource (UCI or Kaggle), respectively. The third and fourth columns represent the size of the instance. The next column contains the best primal bound obtained for each instance. The best dual bound obtained by our solution framework is reported in the sixth column, and the solution time (in seconds) to obtain that bound is shown in the last column.

Table 1: Computational results for penalty function parameters $(\lambda, \gamma) = (1, 3)$

| Datasets | | | | | | |
|---|---|---|---|---|---|---|
| Name | Resource | $p$ | $n$ | Primal bound | Dual bound | Time (s) |
| Graduate Admission 2 | kaggle | 7 | 400 | 20.304 | 20.295 | 22.46 |
| Statlog (Landsat Satellite) | UCI | 36 | 4434 | 5854.507 | 5853.978 | 49.29 |
| Connectionist Bench (Sonar, Mines vs. Rocks) | UCI | 60 | 207 | 71.853 | 68.266 | 195.17 |
| Communities and Crime | UCI | 127 | 123 | 9.891 | 9.466 | 76.44 |
| Urban Land Cover | UCI | 147 | 168 | 1.002 | 0.999 | 5.10 |
| Relative location of CT slices on axial axis | UCI | 384 | 4000 | 10730.592 | 10426.985 | 281.718 |

Table 2: Computational results for penalty function parameters $(\lambda, \gamma) = (10, 30)$

| Datasets | | | | | | |
|---|---|---|---|---|---|---|
| Name | Resource | $p$ | $n$ | Primal bound | Dual bound | Time (s) |
| Graduate Admission 2 | kaggle | 7 | 400 | 22.234 | 22.125 | 8.97 |
| Statlog (Landsat Satellite) | UCI | 36 | 4434 | 5859.670 | 5855.088 | 60.42 |
| Connectionist Bench (Sonar, Mines vs. Rocks) | UCI | 60 | 207 | 110.730 | 105.351 | 18.207 |
| Communities and Crime | UCI | 127 | 123 | 18.456 | 17.560 | 133.35 |
| Urban Land Cover | UCI | 147 | 168 | 10.002 | 9.990 | 8.87 |
| Relative location of CT slices on axial axis | UCI | 384 | 4000 | 16280.696 | 16032.315 | 276.858 |

