# OpenReview forum: "A graph-based global optimization framework for problems with nonconvex norm constraints and penalty functions"
_ICLR.cc/2025/Conference — Submitted to ICLR 2025_

### Official Review · Reviewer_CcjE · 2024-10-29

**Soundness:** 3
**Presentation:** 2
**Contribution:** 2
**Rating:** 5
**Confidence:** 3

**Summary:**

The paper considers convex optimization with a norm bounding constraint. These constraints can be any $p$-norm, where $p \in [0,\infty)$ as well as some non convex penalty terms such as SCAD and MCP. When $p\in (0,1)$, the norm is neither convex nor concave, and as a result standard methods don’t work directly. These kinds of problems appear in several applications and sometimes also appears as a regularizer. Current methods that give exact solutions either use extra variables, thus increasing the size of the problem by a lot, are tailored to the objective function, or are heuristic methods.

This paper gives a unified algorithm that can handle all the different constraints, including non continuous and non convex constraints. The method is based on using decision diagrams. The method roughly uses a scale function to capture these domain constraints, and then uses a decision diagram based relaxation to capture these constraints. The key is that the scale function is able to capture all the listed domain constraints. The paper focusses on proving how to use these scale functions within DD’s to capture the constraints. In the end there are some experiments that use a quadratic function with the SCAD penalty function and they show the solution times for selected datasets with different penalty functions.

**Strengths:**

The paper is able to handle an important set of problems and gives a new technique. It is an interesting connection between non convex optimization and graph theory, since most of the graph theoretic optimization results are captured via convex optimization.

**Weaknesses:**

I think there is no provable bound on the number of iterations, or the rate of convergence of the method. It seems like the authors can prove that the method converges in the limit but there is no way to show the rate of convergence even for simple objectives. Also, in the experiments it is not clear what the method is compared against.

**Questions:**

I have some questions to the authors:

1. Points (i) to (viii) on Pages 2 and 3 seem to have a lot of redundant statements and is repeating what was stated in the paragraph before. I think it can be compressed so that the main message is clear while covering all the points.
2. The SCAD and MCP penalty terms are defined in Prop 1. Would be useful to define in the main text!
3. What all objective functions can be handled by the method? It would be useful to include this explicitly. The constraints are very clear but it is useful to know the functions we can handle as well.
4. Is it possible to show any rates of convergence even for some easy objectives?

---

> ### Author Response · Authors · 2024-11-12
>
> For global solution algorithms that employ divide-and-conquer techniques, it is impossible to provide a bound on the number of iterations—this is a well-known fact common to all general-purpose global algorithms, not specific to our approach. As for comparison with existing methods, we reiterate that no existing methods can globally solve the problem classes examined in our work. Consequently, there are no alternative methods to benchmark our approach against. The breakthrough of our paper is precisely this: for problems previously unsolvable from a global perspective, we present an algorithm capable of achieving global solutions for the first time.

---

> > ### Comment · Reviewer_CcjE · 2024-11-27
> >
> > I have read the authors responses and have decided to maintain my score.

---

### Official Review · Reviewer_HTUi · 2024-11-04

**Soundness:** 2
**Presentation:** 1
**Contribution:** 2
**Rating:** 3
**Confidence:** 3

**Summary:**

The authors propose an innovative graph-based method to globally solve optimization problems with generalized norm-bounding constraints. This approach leverages decision diagrams to create strong convex relaxations for these constraints directly in the original variable space, eliminating the need for auxiliary variables or artificial variable bounds. The preliminary computational experiments on benchmark sparse linear regression problems show the effectiveness of this framework.

**Strengths:**

1. This paper presents a new graph-based framework that leverages decision diagrams to address a broad class of optimization problems with norm-bounding constraints. Unlike traditional methods, it offers a unified approach to solving convex and nonconvex problems.
2. The authors provide mathematical proof showing that, under mild conditions, the proposed method guarantees convergence to global optima.
3. Certain computational experiments are provided on benchmark problems to demonstrate the algorithm's performance.

**Weaknesses:**

1. The proof is unreadable for readers.
2. In Section 4.2, there is no brief description of the method to obtain dual bounds inside an outer approximation framework.
3. In Figure 1, the instance number is difficult for readers to understand.
4. The experiments are insufficient to support your conclusion.

**Questions:**

1. Could you present the proofs of your propositions in an easier way for readers to understand? For instance, consider breaking them into multiple paragraphs instead of a single block of text.
2. In Section 4.2, you should provide a brief description within the main body rather than summarizing it in the appendix.
3. The description of the x-axis in Figure 1 is challenging for readers to understand. Could you clarify its meaning?
4. Can you explain why you chose 5% for a relative optimality gap as the stopping criterion for your algorithm?
5. Did you conduct any experiments to justify setting 50 (and 100) iterations for your tests?
6. Have you conducted any experiments to explore the impact of $\lambda$ and $\gamma$?
7. Have you compared the runtime of your method with the state-of-the-art (SOTA) method for the same problem under either convex or nonconvex conditions?

---

> ### Author Response · Authors · 2024-11-12
>
> 1. The proofs provided are standard and should be straightforward for readers within the relevant fields to follow.
>
> 2. Solving an outer approximation of a problem yields a dual bound, a well-known fact in optimization. Furthermore, there is extensive literature that discusses how an outer approximation can be derived from decision diagrams, making a detailed repetition of these arguments unnecessary and redundant in a conference paper with space limitations. As such, we have provided a summary along with relevant references in the appendix for readers who may be less familiar with this background.
>
> 3. Could you clarify which specific claims you found lacking in computational support? We have clearly indicated that there are no existing methods or state-of-the-art approaches that can solve the problem classes examined here. In other words, our algorithm is the first in the literature capable of globally solving these problems, so direct comparisons with non-existent state-of-the-art methods are not possible. Our claim, mathematically proven, is that we can solve these problems globally, independent of experimental validation.

---

> > ### Comment · Reviewer_HTUi · 2024-11-13
> >
> > 1. The format of your proof is unfriendly to readers.
> > 2. Experiment section: you should refer to questions 5-7.
> > For question 7, my point is: For a convex problem, have you compared your results with any SOTA method? Separately, for a nonconvex problem, have you compared your results with any SOTA method? The intent is not to compare with other SOTA methods across both convex and nonconvex conditions.
> > 3. You didn't answer my "questions".

---

> > > ### Author Response · Authors · 2024-11-13
> > >
> > > We have specifically addressed the "weakness" points raised in the review, which appear to have led to both the questions and the overall rating, rather than the questions themselves, which appear relatively marginal by comparison.
> > >
> > > Regarding the first point, as previously noted, the proof format we used is standard in this field. Breaking down such a standard proof into simpler components would require significantly more space, which is an impractical approach for a conference paper with strict length limitations.
> > >
> > > Regarding the second point, questions 5-7 do not substantiate "weakness #4," which states that "the experiments are insufficient to support your conclusion." We seek clarification on which specific conclusion is deemed unsupported by the experiments. Nowhere do we claim runtime comparisons between our method and state-of-the-art methods for convex or nonconvex cases. Our primary claims are that (a) the proposed framework can globally solve a broad class of problems, which we have mathematically proven; and (b) there are problem structures, such as those involving the SCAD penalty, that cannot be handled by any existing methods. This is due to the fact that SCAD penalties cannot be represented within existing solvers, regardless of parameter selection. In contrast, our framework is mathematically proven to converge to the global optimal solution of the SCAD problem, independent of parameter settings. Therefore, altering parameters would not affect these core conclusions.
> > >
> > > In summary, adding test cases with different parameter settings, iteration counts, or gap tolerance would neither strengthen nor impact the validity of our main conclusions or contributions. The specific test instances and parameters presented are intended solely to demonstrate that our proposed approach is capable of solving these instances, something no other method currently achieves.

---

> > > > ### Comment · Reviewer_HTUi · 2024-11-27
> > > >
> > > > The proof of the theorem can be included in the appendix.
> > > >
> > > > The details provided for the experiments are insufficient and need to be expanded.

---

### Official Review · Reviewer_m5Gj · 2024-11-04

**Soundness:** 3
**Presentation:** 2
**Contribution:** 2
**Rating:** 3
**Confidence:** 2

**Summary:**

The paper introduces a graph-based global optimization framework targeting problems with norm-bounding constraints and nonconvex penalty functions, such as SCAD and MCP. Utilizing decision diagrams (DDs), the method constructs strong convex relaxations directly in the original variable space without adding auxiliary variables. Preliminary experiments on sparse linear regression benchmarks demonstrate the framework’s ability to solve challenging instances that existing global solvers cannot handle.

**Strengths:**

1. The framework accommodates a wide range of norm types, including both convex ($\ell_p$ for $p \geq 1$) and nonconvex ($\ell_p$ for $p \in (0,1)$, SCAD, MCP) regularization, enhancing its versatility across various optimization problems.

2. The paper provides the theoretical convergence guarantee to the global optimum and properties of the convex relaxations generated by the DD-based method.

**Weaknesses:**

1. The computational experiments presented are preliminary and focus on small to medium-sized instances. To fully assess the framework’s scalability and performance, more extensive benchmarking on larger and more diverse datasets is necessary.

2. The DD-based method can grow exponentially with the number of variables. Although node-merging techniques and width limitations are proposed to control DD size, the practical scalability for very large-scale problems remains uncertain and requires further empirical validation.


3. The paper allocates a disproportionate amount of space to providing proofs in the main text while omitting essential technical details, such as the graph-based convexification method.

**Questions:**

Please refer to Weaknesses.

---

> ### Author Response · Authors · 2024-11-12
>
> The main point of this paper is missed in this reviewer's comments. The key point here is that no existing methods can globally solve even small instances of this problem class. Therefore, our ability to achieve global solutions—even for small and medium-sized instances—marks a significant breakthrough in both the theoretical and practical aspects of solving these extremely challenging problems.

---

### Meta-Review · Area_Chair_4brP · 2024-12-21

**Metareview:**

This paper introduces a graph-based optimization framework for norm-bounded and nonconvex norm constraints. The reviewers raised several concerns regarding the clarity of the theoretical results, the scalability of the approach to larger problems, and the insufficient experimental validation of the method. The authors did not provide convincing responses to address these concerns. Overall, the paper requires significant revisions, particularly in improving the readability and presentation of the theoretical results, addressing scalability issues, and enhancing experimental validation of the method. Therefore, I recommend rejecting this paper in its current form.

**Additional Comments On Reviewer Discussion:**

During the rebuttal phase, the authors failed to provide convincing answers and address reviewer m5Gj’s concerns regarding the scalability of the method to larger problems. Moreover, rather than maintaining a constructive attitude and addressing reviewer HTUi’s concern about the readability of the proof, the authors simply asserted that it should not be an issue. Finally, in response to reviewer CcjE’s request for a bound on the number of iterations (or a rate of convergence) and an experimental comparison with existing algorithms, they did not present a convincing answer. Therefore, the reviewers maintained their initial low scores after the rebuttal phase.

---

### Decision · Program_Chairs · 2025-01-22

Reject